

# Seasonal thermal regime and climatic trends in lakes of Tibetan Highlands

Georgiy Kirillin[1], Lijuan Wen[2], Tom Shatwell[1]

[1]Department of Ecohydrology, Leibniz-Institute of Freshwater Ecology and Inland Fisheries (IGB). Berlin, 12587 Germany

[2]Key Laboratory of Land Surface Process and Climate Change in Cold and Arid regions, Northwest Institute of Eco-Environment and Resources, Chinese Academy of Sciences, Lanzhou , 73000 China

*Correspondence to*: Georgiy Kirillin (kirillin@igb-berlin.de)

**Abstract.** The hydrology of the lake-rich Tibetan Plateau is important for the global climate yet little is known about the thermal regime of Tibetan lakes due to scant data. We (i) investigated the characteristic seasonal temperature patterns and recent trends in thermal and stratification regime of lakes on the Tibetan Plateau and (ii) tested the performance of the one-dimensional lake parameterization scheme FLake for the Tibetan lake system. For this purpose we combined three years of in situ lake temperature measurements, several decades of satellite observations and the global reanalysis data. We chose the two largest freshwater Tibetan lakes—Ngoring and Gyaring— as study sites. The lake model FLake faithfully reproduced the specific features of the high-altitude lakes and was subsequently applied to reconstruct the vertically resolved heat transport in both lakes during the last four decades. The model suggested Ngoring and Gyaring were ice-covered for about 6 months and stratified in summer for about 4 months per year with a short spring overturn and longer autumn overturn. In summer the surface mixed boundary layer extended to 6 – 8 m depth and was about 20% shallower in the more turbid Gyaring. The thermal regime of transparent Ngoring responded more strongly to atmospheric forcing than Gyaring, where the higher turbidity dampened the response. According to reanalysis data, air temperatures and humidity increased, whereas solar radiation decreased since the 1970s. Surprisingly, mean lake temperatures did not change, nor did the phenology of ice cover or stratification. Lake surface temperatures in summer increased only marginally. The reason is that the increase in air temperature was offset by the decrease in radiation, probably due to increasing humidity. This study demonstrates that air temperature trends are not directly coupled to lake temperatures and underscores the importance of short-wave radiation for the thermal regime of high-altitude lakes.

## 1 Introduction

The hydrological regime of the Tibetan highlands is extremely complex and highly sensitive to climate changes (Gu et al. 2005; Ma et al. 2012; Yang et al. 2014). The Plateau of Tibet plays a crucial role in the global water cycle because it directly affects the monsoon system and is the origin of major Asian rivers (Su et al. 2015) including the Yellow, Yangtze, Mekong,





Salween, Brahmaputra (Yarlung Tsangpo), and Indus Rivers (Fig. 1). The Plateau contains thousands of lakes, whose hydrological regime is closely connected to glaciers, permafrost, as well as in- and effluents (Zheng 2011; Zhang et al. 2013). The particular climatic environment of the Tibetan Plateau, with low air pressure and intense solar radiation, creates a unique land-atmosphere interaction (Ma et al. 2009), where the lake system is a crucial component of the regional heat and mass

balance (Liu et al. 2009; Gerken et al. 2013; Li et al. 2015; Wen et al. 2015; Dai et al. 2016). Therefore, changes in the regional water budget of the Tibetan Plateau driven by global warming are of key importance for climatic changes on continental scales. Recent evidence indicates that the number of lakes on the Tibetan Plateau is increasing, highlighting global warming effects and local anthropogenic activity, such as infrastructure construction on permafrost (Cheng and Wu 2007; Liu et al. 2009; Yang et al. 2010). Hence, to estimate how lakes may affect the regional climate, it is necessary to quantify the seasonal thermal

dynamics of lakes and ponds, with particular reference to the vertical thermal stratification, the ice regime, and the lake interactions with the lower atmosphere.

Due to the importance of lakes for understanding the global-scale changes on the Tibetan Plateau, several studies have been performed in recent years on lake dynamics and lake-atmosphere interactions. The amount of evaporation and heat flux, and their change, were revealed in lakes in the Tibetan Plateau during the pre- and post-monsoon periods (Haginoya et al. 2009;

Xu et al. 2009). Biermann et al. (2014) measured the turbulent fluxes over wet grassland and a shallow lake with the eddy covariance method at the shoreline in the Lake Nam Co basin. Li et al. (2015) recently studied the characteristics of the energy flux and the radiation balance over Ngoring Lake. With the development of lake schemes and atmospheric models that include the lake surface (Sun et al. 2007), other studies have investigated the effects of the Tibetan Plateau lakes on local climate (Li et al. 2009; Wen et al. 2015; Yang et al. 2015). Nevertheless, compared to other lake-rich regions of the world, like

Fennoscandia, Northern Canada, or the East-Siberian Tundra, the lakes of the Tibetan Plateau are poorly studied, despite their extraordinary thermal regimes due to strong seasonal and synoptic variations.

A major problem in resolving the energy and moisture budget of high-altitude lakes is the lack of long-term observations. Observational data on the physics of Tibetan Plateau lakes are scarce and mostly confined to remote sensing data on lake surface characteristics, such as lake surface temperature (Lin et al. 2011; Zhang et al. 2014). Observations of air-lake fluxes

using eddy covariance methods over Tibetan Plateau lakes are too fragmentary to estimate the seasonal variations (Biermann et al. 2014; Li et al. 2015). Furthermore, lack of knowledge of the heat transport within the lake water column, including seasonal formation of thermal stratification, and thermal dynamics under ice cover, have limited our understanding of lake-atmosphere interactions on climatic scales.

While first brief reports were recently presented (Wang et al. 2014; Wen et al. 2016) on the mixing conditions and vertical

heat transport within the lake water bodies, there are no observational data covering the annual variability or long-term trends in thermal conditions and stratification in Tibetan lakes since no regular lake monitoring was performed in this area in the past. In this situation, modeling of decadal variability in lake dynamics can provide an insight into the ongoing changes on seasonal to climatic time scales, provided that model performance is tested against available observational data on shorter periods. On the other hand, validation of models for parameterizing lakes in weather prediction and climate studies is of significant value





per se, particularly because it is crucial to properly account for the heat budget of one of the largest lake systems worldwide to adequately model heat and mass transport on a planetary level.

In view of these challenges, the present study aimed to:

- Characterize the seasonal stratification regime and ice cover formation in lakes of the Tibetan Plateau using the two largest
freshwater lakes of the region as examples.

- Test the ability of the lake model FLake (Mironov 2008; Kirillin et al. 2011) to simulate the main features of the lake thermodynamics in high-altitude conditions as a potential tool to parameterize the Tibetan lake system in regional climate modeling.

- Quantify the mean trends in the thermal and ice regime of Tibetan lakes during the last half-century and to estimate the
10 climate-driven changes in the seasonal stratification pattern.

For this purpose we combine 3-year observations on the surface heat budget from Ngoring Lake (Wen et al. 2016) with the surface temperatures from the AVHRR SST product (Casey et al. 2010), the one-dimensional lake temperature and mixing model FLake and the long-term forcing from the ERA-Interim and NCEP/NCAR global atmospheric reanalyses (Kalnay et al. 1996; Dee et al. 2011). In situ and satellite data on the surface temperatures are primarily used to validate the model
performance at short and long time scales, respectively. The model is forced by reanalysis data in two different configurations, corresponding to the two neighboring large freshwater lakes located in the north-eastern part of the Plateau—Ngoring and Gyaring. Both lakes undergo virtually the same atmospheric forcing, but differ in mean depth and water transparency. Therefore, a comparison of the model results allows not only quantifying the seasonal and climatic characteristics of the mixing regime, but also estimating the response of the lake thermal structure to variations in the subsurface radiation regime and total
depth of the water column.

## 2 Methods

### 2.1 Study sites

Ngoring (Big Blue Lake) and Gyaring (Big Gray Lake), sometimes referred as the 'twin lakes', are two large freshwater lakes in the source region of the Yellow River (Huang He) on the eastern Tibetan Plateau (Figure 1). The lakes are located at 34.5-
25 35.5°N and 97-98°E, less than 50 km from each other, at an altitude of ~4300 m a. s. l., which counts them among the world's highest freshwater lakes. The Yellow River flows through both Gyaring and Ngoring. The lakes are surrounded by hills covered with alpine meadows. Cold semi-arid continental climate prevails in the lake basin, and the long-term (1953–2012) monthly mean air temperature varies from 7.7 °C in July to -16.2 °C in January, with an annual mean of -3.7 °C (Li et al., 2015); the average annual precipitation is 321.4 mm (Data from China's National Climate Center). The lakes are ice-covered from early
December to mid-April.

Ngoring Lake has a surface area of 610 km$^2$. The mean and maximum depths are 17 and 32 m, respectively. The mineralization in the lake water is low with a conductivity of ~0.4 mS cm$^{-1}$. The lake is oligotrophic, fish in the lake are rare, and aquatic





plants grow only in the riparian areas. The reported lake water transparency does not exceed 3m (Kar 2014). Gyaring Lake has a similar surface area (526 km$^2$) but is shallower than Ngoring Lake with an average depth of 8.9 m and maximum depth of 13.1 m. Lake water is more turbid than in Ngoring, with Secchi depth (transparency) of only 1 m (Kar 2014). Due to the close geographical location and similar surface areas, both lakes are subject to virtually the same atmospheric forcing, but differ in

their depths and water transparency – two major factors determining radiative heat transport within the water column and formation of thermal stratification. Also, both lakes are fresh, so that effects of salt on density stratification can be neglected in modeling.

### 2.2 Lake temperature data

The lake-atmosphere heat exchange and characteristics of the air-lake boundary layer were systematically observed over

Ngoring Lake during the summer open water seasons of 2011-2013 (Li et al. 2015; Wen et al. 2015; Li et al. 2016; Wen et al. 2016). The bulk surface temperatures obtained in these observations comprise the primary dataset for verifying the lake model in this study. The bulk temperatures of the lake surface water were recorded by 109L probes (Campbell Scientific, Inc.) at water depths of 0.05, 0.20, 0.40 and 0.60 m during the observation period. All sensors in the 0.6 m thick surface layer recorded essentially the same temperatures within the probes' accuracy, so the 4 time series were vertically averaged for the subsequent

analysis. The incoming and outgoing shortwave and longwave radiation were measured with a net radiometer (CNR-1/CNR-4, Kipp and Zonen) 1.5 m above the lake.

### 2.3 Modeling

Modeling was performed with the lake temperature and mixing model FLake  (Mironov 2008; Kirillin et al. 2011)—a highly parameterized bulk model of the lake thermal regime specifically designed to parameterize inland waters in regional climate

models and numerical weather prediction. FLake is based on a two-layer parametric representation of the vertical temperature structure. The upper layer is treated as well-mixed and vertically homogeneous. The structure of the lower stably-stratified layer, the lake thermocline, is parameterized using a self-similar representation of the temperature profile. The same self-similarity concept is used to describe the temperature structure of the thermally active upper layer of the bottom sediments (Golosov and Kirillin 2010) and the ice cover (Mironov et al. 2012). The depth of the mixed layer is computed from the

prognostic entrainment equation in convective conditions, and from the diagnostic equilibrium boundary-layer depth formulation in conditions of wind mixing against the stabilizing surface buoyancy flux. The integrated approach implemented in FLake allowed combining high computational efficiency with realistic representation of the major physics behind turbulent and diffusive heat exchange in the stratified water column. As a result, FLake is widely used for lake representation in land schemes of regional climate models, being implemented, among others, into the surface schemes of the Weather Research and

Forecasting model (WRF, Mallard et al. 2014), HTESSEL (ECMWF, Dutra et al. 2010), SURFEX (Meteo France, Salgado and Le Moigne 2010), and JULES (UK Met Office, Rooney and Jones 2010). Thanks to its robustness and computational efficiency, FLake has become a standard choice in climate studies involving the feedbacks between inland waters and the



atmosphere, and is used operationally in the NWP models of the German Weather Service (DWD), the European Centre for Medium-Range Weather Forecasts (ECMWF), the UK Met Office, the Swedish Meteorological and Hydrological Institute (SMHI), the Finnish Meteorological Institute (FMI) and others.

### 2.4 Lake temperatures from SST Pathfinder product

As an additional source of the lake surface temperature estimations, we used AVHRR Pathfinder Version 5.2 (PFV5.2) remote sensing data, obtained from the US National Oceanographic Data Center and GHRSST (http://pathfinder.nodc.noaa.gov). The PFV5.2 data are an updated version of the Pathfinder Version 5.0 and 5.1 collection described in Casey et al. (2010). Among the available lake surface temperature products, the AVHRR Pathfinder data have the longest time coverage (1982-2012) and a 4 km spatial resolution, distinguishing both Gyaring and Ngoring as water surfaces. The lake mask of the PFV5.2 includes
25 grid points for Gyaring, 5 of which are 'open water' points not adjoining the lake shoreline, and 31 points for Ngoring, 8 of which are 'open water' points. In the further analysis, only spatial averages of water temperatures from the open water points were used (Figure 1) to reduce the potential influence of the surrounding land on the temperature readings.

### 2.5 Long-term model forcing from reanalysis data

In the absence of direct climatic observations for the study area, we adopted two widely used global reanalyses of atmospheric
fields, the NCEP/NCAR Reanalysis 1 (previously NCEP/NCAR Reanalysis 40, Kalnay et al. 1996), and the ERA Interim Reanalysis produced by the European Centre for Medium-Range Weather Forecasts (ECMWF, Dee et al. 2011). Both datasets rely on different atmospheric models and data assimilation schemes. Accordingly, we first compared the surface layer meteorology of the Tibetan Plateau from the two reanalysis datasets, and then we investigated how the surface meteorology affected the modeled lake physics. The parameters used to force the lake model were 6-hour surface layer characteristics: 2-m
air temperature $T_a$ [recalculated to °C], 2-m air humidity $e_a$ [mb], incoming solar radiation $I_R$ [W m$^{-2}$], 10-m wind speed $U_{10}$ [m s$^{-1}$], and total cloud cover $N$ [0-1]. The period 1975-2014 was applied for modeling forced by NCEP/NCAR, and a shorter period of 1979-2014 was used for the ERA-driven modeling, since the ERA Interim data are available since 1979.

### 2.6 Model setup and validation

We avoided tuning the model-specific parameters and algorithms to the observed lake data, with the exception of the lake heat
content at the initial stage of the open water season. The water temperature immediately after the ice breakup was set to 4°C independently of the modeled water temperatures during the preceding ice cover period. This amendment aimed to account for heating of the water column by solar radiation during the ice covered period, which is not considered in the current version of FLake. Neglecting heating by solar radiation penetrating the ice cover did not introduce large errors in FLake applications to low-altitude, seasonally ice-covered lakes (Kirillin 2010; Bernhardt et al. 2012) where the incoming solar radiation in winter
is lower and the relatively thick snow cover dampens the radiative heating. However, in the high-altitude, low-latitude lakes




of the Tibetan Plateau, the radiation penetrating the ice appears to be an important component of the lake heat budget, so that the water temperatures directly after ice breakup are close to the maximum density value (Wen et al. 2016).

The water transparency characteristics used in the model configuration were adopted from (Kar 2014). The light extinction coefficient $\gamma$ was calculated as the reciprocal of the Secchi depth $h_S$, $\gamma = C_\gamma h_S^{-1}$, assuming $C_\gamma = 1.8$, an approximate mid-value of the range reported in previous studies (see reviews in Kirillin and Shatwell 2016 and Shatwell et al. 2016).

Model performance was assessed by comparing daily averages of modeled and observed (in situ and satellite) surface temperature ($T_s$) between May and October. The satellite data contained mostly only one data point on a given day, which may increase error. Some outliers existed usually under ice cover or subzero skin temperatures. All $T_s \leq 0$ were excluded from modeled and observed data, i.e. also the whole ice covered period. Different aspects of model performance were examined including model bias (Eq. 1), centred root mean square error ($RMSE_c$, Eq. 2), normalized standard deviation ($\sigma_{norm}$, Eq. 3), as well as normalized root mean square error ($I_2$ and $I_3$, Eq. 4 and 5).

$$bias = \bar{m} - \bar{o} \tag{1}$$

$$RMSE_c = \sqrt{\frac{1}{n}\sum_{i=1}^n ((m_i - \bar{m}) - (o_i - \bar{o}))^2} \tag{2}$$

$$\sigma_{norm} = \sqrt{\sum_{i=1}^n (m_i - \bar{m})^2 / \sum_{i=1}^n (o_i - \bar{o})^2} \tag{3}$$

$$I_2 = \sqrt{\frac{1}{n}\sum_{i=1}^n (o_i - m_i)^2} \Big/ \sqrt{\frac{1}{n}\sum_{i=1}^n o_i^2} \tag{4}$$

$$I_3 = \sqrt{\frac{1}{n}\sum_{i=1}^n (o_i - m_i)^2} \Big/ \sqrt{\frac{1}{n}\sum_{i=1}^n (o_i - \bar{o})^2} \tag{5}$$

Here $m_i$ and $o_i$ are the modeled and observed values, respectively, and $\bar{m}$ and $\bar{o}$ are their means.

**2.7 Statistics and trend analysis.**

The lakes were defined as stratified when the difference between surface and bottom temperature, $T_s - T_b > 0.5$ K for summer stratification, and $T_s - T_b < -0.5$ K or when the ice cover thickness $h_{ice} > 0$ for winter stratification. Stratification duration, as well as stratification start and end dates, given in day of the year (doy), refer to the longest uninterrupted stratification event each year/season. The lakes were defined as mixed when they were not stratified. Spring and autumn overturn were defined as the longest uninterrupted period of mixing following winter and summer stratification, respectively. In simulations, when the lakes froze in winter, they always remained frozen until thaw in spring. Therefore, the ice cover duration, as well as freeze and thaw dates, refer to both the longest uninterrupted period of ice cover and the total number of days of ice cover in each winter season. Seasons are defined from typical hydrological and ice regime as winter: Jan-Mar, Spring: Apr – Jun, Summer: Jul-Sep, Autumn: Oct-Dec. Trends in data were assessed using linear regressions. Homogeneity of variance and normality of residuals were assessed by inspecting plots of residuals vs fitted values and normal quantile-quantile plots. Leverage was assessed using Cook's distances. All statistical analyses were performed with R version 3.3.0 (R Core Team 2016).



## 3. Results

### 3.1 Model performance

The model simulated the lake surface temperatures during the ice-free period well (Fig. 2, Table 1). The temperature bias of the model was negligible (<1% of the range of temperature variability) forced by both the NCEP and ERA reanalyses (Fig. 4).

The correlation between modeled and observed values was notably high for the ERA input: $r = 0.96$, or 92% of variance captured by the model. The NCEP forcing provided slightly worse results: $r = 0.93$ or 86% of correctly simulated variance. The normalized standard deviation ($\sigma_{norm}$) was also better for the ERA forcing, differing from 1 by less than 0.3%, whereas $\sigma_{norm}$ was around 1.25 for the NCEP input (Fig. 4).

Comparison of the model results to the satellite data demonstrated a slightly worse but still qualitatively acceptable agreement

(Table 1, Fig. 3). Among obvious reasons for the disagreement is the low and irregular temporal resolution of the remote sensing data, as well as the differences between the surface temperatures in situ and the values derived empirically from remote infrared radiation measurements. The validation of the satellite data against the bulk in situ temperatures yielded similar divergences to the model-satellite comparison, indicating that the uncertainties in the satellite data were the primary reason for their deviation from the modeling results. The satellite data from the Pathfinder SST produced generally higher surface

temperatures than both in situ data and modeling results, with the bias varying between 0.4 K for the three years of observations and 0.7-1.5 K for the 30 years of the modeling results (Table 1).

The strongest deviations of the model results from the in situ temperatures and remote sensing data were observed in the late spring and early summer (Figs. 2, 3). These deviations resulted apparently from the model uncertainties in the prediction of the ice cover breakup. No long-term time series on the ice breakup dates are available for the lakes of the region, making it

impossible to quantify the model performance on the ice regime. An indirect estimation of the ice breakup dates from the surface temperatures in 2011-2013 suggests that the model error amounts to several weeks.

Since the model driven by the ERA data performed generally better than with the NCEP input, the results on seasonal characteristics and climatic trends are discussed below based on the ERA-FLake configuration. The outcomes of the NCEP-driven model were overall similar to those of the ERA-driven model with slightly higher water temperatures and weaker

thermal stratification. The statistics are presented in Table 2, and an extended discussion of NCEP results is omitted for the sake of brevity.

### 3.2 Seasonal temperature and mixing regime

The modeled seasonal course of thermal stratification and ice regime was typical for a dimictic regime (Hutchinson 1975; Kirillin and Shatwell 2016) in both Tibetan lakes (Table 2). The ice-covered period lasted for 26-29 weeks and the ice thickness

reached its maximum of 0.8-1.0 m in early March. Ice melted on 16 May +-1 week, followed by a 2-3 week period of full mixing (spring overturn). These phenological characteristics of the seasonal stratification were similar in Ngoring and Gyaring since they were not significantly affected by the differences in the morphology and water transparency. Both lakes stratified





for 16-19 weeks in summer (Fig. 5). The autumn overturn (period of complete mixing) started around 14 Oct in Ngoring and about 3 weeks earlier in the shallower Gyaring. Autumn mixing also lasted longer in Gyaring, so that summer stratification was shorter. The full mixing period lasted 4-6 weeks, which is two times longer than the spring overturn - a remarkable feature of the seasonal mixing regime with important effects on the lake biogeochemistry as discussed below. The summer surface

temperatures in Ngoring (12.01 °C) and Gyaring (12.16 °C) were similar. The temperatures at the lake bottom were in turn appreciably higher in shallower Gyaring (7.96 °C vs. 6.26 °C), which also implies generally higher mean temperatures in this lake. The differences in the water transparency between Ngoring and Gyaring were evident primarily in the thickness of the surface mixed layer $h_{mix}$ (epilimnion) in summer: in Ngoring $h_{mix}$ was 1.5-2.0 m deeper than in Gyaring (Fig. 6).

### 3.3 Trends in atmospheric forcing

*Atmospheric forcing*: The mean trends in the principal atmospheric variables differed in the NCEP and ERA reanalyses (Table 3). The ERA data showed significant trends only in air temperature (as warming at ~0.28°C per decade) and in air humidity. A positive humidity trend was present in the NCEP data too, whereas no trend existed in the air temperature. In turn, the NCEP data suggested a weak but statistically significant increase of cloud cover and a barely distinguishable decrease in wind speeds significant at ~97 % confidence level. The picture was clearer when the trends were split into seasonal bins: both reanalyses

showed an increase in air temperatures in summer of 0.2-0.5°C per decade (with higher trend values in the ERA data). $T_a$, however, remained unchanged in spring and sank in autumn (statistically significant in NCEP only, at -0.5 °C per decade, and insignificant in ERA at -0.15 °C per decade). The most significant increase of $T_a$ occurred in winter (slightly stronger in ERA at 0.7°C per decade), resulting in a positive trend in annual $T_a$ in the ERA data, but an insignificant trend in annual $T_a$ in the NCEP data. The decrease in the solar radiation in summer, as the major driver of the regional heat budget, was remarkable in

both datasets (-3.8 W m$^{-2}$ decade$^{-1}$ in ERA vs -1.2 W m$^{-2}$ decade$^{-1}$ in NCEP).

### 3.4 Long-term lake response to atmospheric trends

Several characteristics of the seasonal thermal regime did not reveal any significant long-term trends in either of the two lakes based on model simulations with both reanalysis datasets. These characteristics included the dates of onset and breakup of the ice cover, ice thickness, dates of full mixing in spring, dates of the maximum ice thickness and of the maximum surface

temperature, bottom temperature and mean lake temperatures in summer. The fact that mean water temperature remained essentially unaffected by long-term atmospheric trends in both lakes is rather unexpected in view of the numerous reports on lake warming throughout the world. In the ERA-driven results, the surface temperature, $T_s$, did not increase significantly in either lake, whereas according to the NCEP reanalysis, $T_s$ increased at 0.24°C per decade and 0.18°C per decade in Ngoring and Gyaring, respectively (Table 4).

There were marginally insignificant signs of an increase in stratification and decrease in mixing across the lake water column in Ngoring, but not in Gyaring, under NCEP forcing, including a weak decrease of the mixed layer depth (0.27 m decade$^{-1}$,





significance level 0.06 only) accompanied by a decrease in the deep water temperatures (0.29 K decade$^{-1}$, p=0.10). Also the increase in $T_s$ was slightly weaker in Gyaring (0.18 K decade$^{-1}$ vs. 0.24 K decade$^{-1}$ in Ngoring).

## 4 Discussion

### 4.1 Model performance

FLake demonstrated good abilities to capture the thermal characteristics of high-altitude lakes. In particular, the good simulation of the surface temperature indicates that the model is suitable for lake representation in coupled land-atmosphere modeling of the Tibetan Plateau and other alpine regions. The performance of the model for the two Tibetan lakes is comparable to that reported previously in temperate (Kirillin 2010; Stepanenko et al. 2014; Shatwell et al. 2016) and tropical (Thiery et al. 2014) lakes. The results underscore the importance of the atmospheric input for the correct simulation of lake

thermodynamics: The ERA Interim—a more recent reanalysis dataset with updated data assimilation—provides an appreciably better FLake output, while the older generation NCEP/NCAR dataset demonstrates an appreciable bias in lake temperatures as well as slightly higher root-mean-square error. A better performance of the ERA reanalyses over the Tibetan highlands, in particular, in reproducing the observed long-term trends in air temperature, has been also reported in earlier studies (Bao and Zhang 2013).

The major drawback of the model, which should be taken into account when simulating the thermal regime of Tibetan lakes, as well as other high-altitude lake systems with an extended ice-covered season and low snow precipitation in winter, is an oversimplified representation of the winter conditions during the ice-covered period. The performance of FLake at simulating ice cover in temperate freshwater lakes was previously analyzed by Bernhardt et al. (2012). FLake, as well as all other lake parameterization schemes currently used in land-atmosphere coupling of climate models (Hostetler et al. 1993; Subin et al.

2012), neglects heating of the water column by solar radiation penetrating the ice cover. One of the negative effects of this simplification is the error in simulating the ice thickness and the ice breakup date. It was not possible to quantitatively estimate this effect in the present study due to the lack of long-term observations on ice-on and ice-off dates. Nevertheless, the modeled duration of the ice cover from October-November to May-June and the ~1 m maximal ice thickness in Tibetan lakes agree with the existing occasional field observations (Wen et al. 2016). The modeling results on the ice thickness may be assumed

more reliable than ice break-up dates since ice growth is mainly governed by the ice-atmosphere heat budget (Aslamov et al. 2014; Leppäranta 2015), which is well captured by the model. Another apparent error introduced by neglecting the under-ice radiative heating is that the mean heat content of the lake after ice-off is underestimated. The result is a strong underestimation of the lake surface temperatures in early summer. Taking into account strong constraints on the computational costs of the lake parameterization schemes, a remarkable improvement in regional climate models can be achieved by extending the lake

modules to include one of the integrated bulk algorithms of solar heating and convective mixing developed earlier (Mironov et al. 2002; Oveisy and Boegman 2014).




The surface temperatures provided by the AVHRR Pathfinder product were generally higher than both in situ and modeled temperatures. The reason for the positive bias is not clear: we may tentatively relate the error to an overestimation of the cool skin effect by the current AVHRR algorithm and hypothesize that corrections for the atmospheric radiation, used in the empirical Pathfinder SST algorithms do not perform well over high-mountain regions with low atmospheric density/radiation.

The cool skin effect, while not considered directly in our study, is however expected to be even stronger in high-altitude lakes than in the ocean, due to strong radiative heating and permanent cooling at the lake surface (Li et al. 2015).

### 4.2 Seasonal thermal regime and ice conditions

Both studied lakes are clearly dimictic and share the typical features of this type of mixing regime found in other regions: two periods of weak vertical mixing lasting several months separated by shorter periods of full vertical mixing (overturns) in spring

and in autumn. As follows from the modeling results, a particular difference of the dimictic regime in Tibetan lakes from those in lowland temperate regions is the comparably short period of spring overturn (SO), so that summer stratification forms only about one week after ice-off. This is apparently due to the alpine climatic conditions, where the low air temperatures produce relatively long ice-covered periods, which last until late May or early June. The solar radiation flux in the early summer is however much stronger than at lower altitudes and quickly heats the upper water column directly after ice-off.

Since spring overturn was short, the major biogeochemical processes that take place when oxygenated surface water comes into contact with the highly mineralized sediment should occur in autumn, when overturn is longer. The characteristics of biogeochemical processes in Tibetan lakes are largely unknown, but some key features, in particular, the oxygen regime, may be deduced from the seasonal stratification pattern (Golosov et al. 2007). Due to the low eutrophication of both lakes, oxygen consumption rates are probably low.  Strong vertical oxygen gradients and an oxygen deficit in the deep parts of the lakes

probably develop during ice-covered periods, when the lake surface is isolated from the atmosphere for several months. This oxygen deficit in turn may trigger biogeochemical processes at the interface between the water and the highly mineralized lake sediment, such as methane production. The spring overturn may be too short to completely replenish the depleted oxygen after winter, so that the anoxic conditions may persist in the deep layers of lakes throughout the summer season, isolated by the stratification. In this case, the most intensive exchange between the lake hypolimnion and the atmosphere, including potential

emissions of methane and carbon dioxide should most probably appear during the autumn overturn.

### 4.3 Lake response to climatic trends in external forcing

Surface temperatures of lakes are recognized as important indicators of regional climate change driven by global warming (Adrian, others). In temperate climate regions, warming trends in lakes are reported to be close to the air temperature trends (Arhonditsis et al. 2004; Danis et al. 2004; Kirillin 2010; Schneider and Hook 2010). Some studies even suggest that temperate

lakes warm faster than the atmosphere due to shortening of the ice covered period and, as a result, higher storage of incoming solar radiation (Austin and Colman 2007; Kintisch 2015). On the other hand, tropical and alpine lakes tend to warm more slowly than the air above (Livingstone 2003; Vollmer et al. 2005; Coats et al. 2006). A recent synthesis of available lake





temperature data (O'Reilly et al. 2015) has revealed high geographic variability in lake warming trends over the globe. In particular, no significant warming trends were determined for lakes on the Tibetan Plateau. Our results indeed suggest that Tibetan lakes have not been warming during the last several decades and provide insight into the mechanisms behind this.

Atmospheric warming trends were present in both reanalysis datasets, although these were unevenly distributed over the seasons. Air temperature increased in summer (more strongly in the ERA reanalysis at 0.5 K decade[-1] than in NCEP at 0.2 K decade[-1]), remained unchanged in spring, decreased in autumn (significantly only in NCEP at -0.5 K decade[-1], and insignificantly in ERA at -0.15 K decade[-1]), and increased in winter (slightly more in ERA at +0.7 K decade[-1] vs +0.4 K decade[-1] in NCEP). The winter warming trend was stronger than the summer trend, but its effect on lakes is weak due to isolation from the atmosphere by the ice cover. The statistically significant but rather small tendency for ice thickness to decrease at ~1 cm decade[-1] simulated in both Ngoring and Gyaring may be related to the overall effect of a warmer atmosphere in winter. Despite the overall warming of the atmosphere, which was especially apparent in the ERA dataset, there was no significant trend in the lake surface temperatures in the ERA-forced results, and only a slight increasing trend in the NCEP results. This fact is closely connected to the negative trend in the solar radiation, which decreased in the ERA dataset at 3.8 Wm[-2] decade[-1], more than 3 times faster than in the NCEP dataset. A characteristic feature of Tibetan lakes as compared with lowland lakes is their stronger heating by solar radiation and, as a result, a persistently warmer lake surface than the atmosphere (Wen et al. 2016). Hence, the effects of the air temperature increase on the lake heat budget were counteracted by the reduced radiative heating, resulting in no significant long term change of the mean lake temperatures (Fig. 7). This core result of the present study indicates that trends in the lake temperatures are not coupled directly to the increase of the air temperature and underscores the crucial role of short-wave radiation for thermal conditions in alpine lakes.

In general, the response of the stratification regime to changes in external atmospheric forcing in deep and clear Ngoring was stronger than in shallow and more turbid Gyaring, which was most apparent in the NCEP-forced results. A plausible explanation of this effect is that the high turbidity of the Gyaring water dampened the effects of higher air temperatures and weaker solar radiation. In the transparent waters of Ngoring, the effect of the radiation decrease is distributed over a water column several meters thick, whereas the increase of the air temperature affects the heat transport at the upper air-water interface. This decoupling of the two counteracting trends produced a temperature increase at the upper boundary with a simultaneous decrease of the heat content in the water column beneath, resulting in a shallower mixed layer and stronger stratification. In Gyaring, on the other hand, both the decrease in radiation and the increase in air temperature are concentrated at the lake surface: the daytime heat is stored in a shallower surface layer, and is more quickly released to the atmosphere by night-time cooling. Thus, the opposing trends in air temperature and radiation effectively cancel each other at the lake surface with little effect on the thermal characteristics of deeper waters.

The radiation decrease can be hypothetically related to the direct effect of anthropogenic aerosols at high altitudes. Another possible reason for the lower solar radiation flux in the Tibetan highlands might be increased evaporation, and as a result, higher air humidity, which is also supported by positive trends in the humidity and cloud amount present in reanalysis data. In particular, Shen et al. (2015) report local cooling over the Tibetan Plateau due to increased evaporation resulting from increased




vegetation growth. The exact mechanisms of the decrease of the solar radiation of the Tibetan Plateau require further investigation.

**Author contribution**

GK conceived and designed the study, and performed modelling and analysis of the remote sensing data. LW performed collection and analysis of field measurements. TS evaluated the model performance and performed statistical trend analysis. All authors contributed to the final analysis of results. GK and TS wrote the ms with contributions from LW.

**Competing interests**

The authors declare that they have no conflict of interest.

**Acknowledgements**

NCEP reanalysis data was provided by the NOAA/OAR/ESRL PSD, Boulder, Colorado, USA from their website at http://www.esrl.noaa.gov.psd/. ERA-Interim data provided courtesy ECMWF. AVHRR data were provided by GHRSST and the US National Oceanographic Data Center. Figure 1 was prepared by David Shatwell. GK was supported by the Opening Research Foundation of Key Laboratory of Land Surface Process and Climate Change in Cold and Arid Regions, Chinese Academy of Sciences (LPCC201401), and the German Science Foundation (DFG Projects KI-853-7/1, KI-853-11/1). LW was supported by the National Natural Science Foundation of China (91637107, 41475011). TS was partially supported by the DFG Project KI-853-7/1.

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





**Tables**

**Table 1: Goodness of fit statistics for simulations in Ngoring and Gyaring based on surface temperatures. Simulations were forced by either the ERA-Interim or the NCEP/NCAR reanalyses. The last row shows a comparison of satellite-derived surface temperatures with in-situ measurements in Ngoring. Statistics are described in the methods (Eq. 1 – 5).**

| Lake | Forcing | Reference data | Bias (K) | *RMSE* (K) | *RMSE$_c$* (K) | *I$_2$* | *I$_3$* | $\sigma_{norm}$ |
|---|---|---|---|---|---|---|---|---|
| **Ngoring** | **NCEP** | **in-situ** | **0.24** | **2.07** | **1.62** | **0.15** | **0.42** | **1.25** |
| **Ngoring** | **ERA** | **in-situ** | **0.11** | **1.11** | **1.10** | **0.10** | **0.29** | **1.00** |
| Ngoring | NCEP | satellite | -0.77 | 2.12 | 1.98 | 0.21 | 0.54 | 1.25 |
| Ngoring | ERA | satellite | -0.70 | 1.87 | 1.73 | 0.18 | 0.48 | 1.03 |
| Gyaring | NCEP | satellite | -1.47 | 3.17 | 2.81 | 0.32 | 0.87 | 1.48 |
| Gyaring | ERA | satellite | -1.42 | 2.47 | 2.01 | 0.25 | 0.67 | 1.22 |
| **Ngoring** | **In situ** | **satellite** | **-0.39** | **2.49** | **2.73** | **0.25** | **0.64** | **0.77** |



Table 2: Mean characteristics of the seasonal thermal regime in Ngoring and Gyaring forced by the NCEP/NCAR and ERA-Interim reanalysis datasets. $T_s$, $T_b$, and $T_m$ are temperatures at the surface, bottom, and averaged over the whole water column, respectively; $h_{mix}$ and $h_{ice}$ are the thickness of the mixed layer and the ice cover respectively; doy is day of the year.

| Variable | Units | Ngoring NCEP | | Ngoring ERA | | Gyaring NCEP | | Gyaring ERA | |
|---|---|---|---|---|---|---|---|---|---|
| | | mean | sd | mean | sd | mean | sd | mean | sd |
| Maximum $T_s$ | °C | 15.6 | 1.1 | 14.1 | 0.8 | 16.5 | 1.2 | 15.1 | 0.9 |
| Timing of maximum $T_s$ | doy | 222.2 | 10.8 | 220.1 | 9.8 | 212.2 | 13.5 | 211.0 | 14.0 |
| Stratification duration | d | 113.6 | 15.9 | 134.6 | 12.9 | 81.6 | 32.2 | 115.5 | 14.6 |
| Stratification start | doy | 165.1 | 10.6 | 153.3 | 8.2 | 167.8 | 15.0 | 151.3 | 8.0 |
| Stratification end | doy | 278.8 | 7.4 | 287.9 | 6.2 | 251.3 | 16.9 | 266.7 | 9.2 |
| Winter stratification duration | d | 196.3 | 6.0 | 180.6 | 10.0 | 203.8 | 7.3 | 186.0 | 9.7 |
| Total Ice duration | d | 189.9 | 5.6 | 173.0 | 9.2 | 200.4 | 6.1 | 182.7 | 9.9 |
| Freeze time | doy | -45.9 | 3.1 | -35.2 | 5.5 | -55.7 | 4.0 | -44.3 | 5.3 |
| Thaw time | doy | 142.6 | 4.9 | 136.5 | 7.6 | 143.2 | 4.9 | 137.0 | 7.5 |
| Maximum $h_{ice}$ | m | 1.0 | 0.04 | 0.8 | 0.04 | 1.0 | 0.04 | 0.9 | 0.04 |
| Timing of maximum $h_{ice}$ | doy | 68.8 | 7.2 | 57.9 | 10.0 | 67.0 | 6.2 | 56.1 | 11.4 |
| Spring overturn duration | d | 22.0 | 11.0 | 16.3 | 9.0 | 20.1 | 12.4 | 13.1 | 8.9 |
| Spring overturn start | doy | 142.6 | 4.9 | 136.5 | 7.6 | 143.7 | 5.3 | 138.8 | 8.3 |
| Spring overturn end | doy | 163.7 | 10.5 | 152.3 | 8.2 | 166.1 | 13.7 | 150.2 | 7.9 |
| Autumn overturn duration | d | 30.9 | 8.9 | 31.1 | 8.6 | 52.2 | 19.5 | 46.9 | 11.4 |
| Autumn overturn start | doy | 279.2 | 7.0 | 288.0 | 6.1 | 251.1 | 18.4 | 267.2 | 10.3 |
| Autumn overturn end | doy | 310.1 | 3.6 | 319.1 | 4.1 | 303.3 | 3.9 | 314.1 | 4.4 |
| $T_s$ (summer mean) | °C | 12.8 | 0.7 | 12.0 | 0.5 | 13.1 | 0.6 | 12.2 | 0.5 |
| $T_m$ (summer mean) | °C | 10.4 | 0.5 | 9.2 | 0.5 | 11.7 | 0.8 | 10.2 | 0.7 |
| $T_b$ (summer mean) | °C | 7.3 | 1.3 | 6.3 | 1.0 | 9.7 | 1.7 | 8.0 | 1.1 |
| $h_{mix}$ (summer mean) | m | 7.5 | 1.0 | 5.6 | 0.5 | 5.6 | 1.2 | 3.8 | 0.6 |
| $h_{ice}$ (winter mean) | m | 0.9 | 0.03 | 0.8 | 0.04 | 0.9 | 0.03 | 0.8 | 0.04 |





**Table 3: Comparison of meteorological variables in the ERA and NCEP reanalysis datasets.** $I_R$ : incoming solar radiation [W m$^{-2}$], $T_a$ : 2-m air temperature [°C], $e_a$ : 2-m air humidity [mb], $U_{10}$ : 10-m wind speed [m s$^{-1}$], $N$ : total cloud cover [0-1]. Trends are given in units of the variable per year when significant at $p < 0.05$ (*), $p < 0.01$ (**) and $p < 0.001$ (***).

| Variable | ERA | | | NCEP | | |
|---|---|---|---|---|---|---|
| | Mean | sd | Trend | Mean | sd | Trend |
| *Annual* | | | | | | |
| $I_R$ | 191.97 | 3.75 | | 179.25 | 1.61 | |
| $T_a$ | -4.47 | 0.63 | **0.0282** | -5.59 | 0.49 | |
| $e_a$ | 3.30 | 0.18 | **0.0085** | 6.34 | 0.38 | **0.0156** |
| $U_{10}$ | 4.01 | 0.11 | | 4.98 | 0.20 | **-0.0060*** |
| $N$ | 0.57 | 0.03 | | 0.32 | 0.02 | **0.0008** |
| *Winter* | | | | | | |
| $I_R$ | 167.63 | 2.35 | | 152.07 | | |
| $T_a$ | -13.11 | 1.49 | **0.0696** | -16.19 | 1.45 | **0.0486*** |
| $e_a$ | 1.09 | 0.11 | | 2.77 | 0.21 | **0.0095*** |
| $U_{10}$ | 5.11 | 0.38 | | 5.72 | | |
| $N$ | 0.58 | 0.05 | | 0.24 | 0.03 | **0.0009*** |
| *Spring* | | | | | | |
| $I_R$ | 247.25 | 6.62 | | 232.41 | 2.84 | |
| $T_a$ | 0.46 | 0.58 | | 0.18 | 0.68 | |
| $e_a$ | 4.03 | 0.28 | | 7.91 | 0.58 | **0.0194*** |
| $U_{10}$ | 3.76 | 0.16 | | 4.96 | 0.27 | **-0.0089*** |
| $N$ | 0.67 | 0.04 | **0.0013*** | 0.42 | 0.03 | |
| *Summer* | | | | | | |
| $I_R$ | 204.80 | 10.26 | **-0.3824*** | 204.39 | 4.36 | **-0.1196*** |
| $T_a$ | 4.74 | 0.84 | **0.0504*** | 6.09 | 0.71 | **0.0248** |
| $e_a$ | 6.41 | 0.48 | **0.0262*** | 11.03 | 1.06 | **0.0314*** |
| $U_{10}$ | 2.98 | 0.13 | | 4.21 | | |
| $N$ | 0.61 | 0.05 | | 0.43 | | |
| *Autumn* | | | | | | |
| $I_R$ | 148.35 | 3.68 | | 128.21 | 1.40 | **0.0468*** |
| $T_a$ | -10.08 | 1.00 | | -12.58 | 1.03 | **-0.0460** |
| $e_a$ | 1.63 | 0.17 | | 3.59 | 0.24 | |
| $U_{10}$ | 4.22 | 0.32 | | 5.03 | 0.35 | |
| $N$ | 0.43 | 0.05 | | 0.22 | 0.03 | **0.0010*** |




Table 4: Trends in characteristics of the lake thermal regime. Abbreviations as for Table 2.

| Variable | Units | Ngoring NCEP | | Ngoring ERA | | Gyaring NCEP | | Gyaring ERA | |
| --- | --- | --- | --- | --- | --- | --- | --- | --- | --- |
| | (per year) | Trend | $p$ | Trend | $p$ | Trend | $p$ | Trend | $p$ |
| Maximum $T_s$ | °C | **0.032\*** | *0.023* | | | | | | |
| Stratification duration | d | **0.492\*** | *0.022* | | | | | | |
| Stratification start | doy | *-0.274* | *0.058* | | | | | | |
| Stratification end | doy | **0.218\*** | *0.029* | | | | | | |
| Winter stratification duration | d | | | | | | | **-0.376\*** | *0.023* |
| Maximum $h_{ice}$ | m | **-0.001\*** | *0.034* | | | **-0.001\*** | *0.031* | **-0.001\*** | *0.037* |
| Autumn overturn duration | d | **-0.338\*\*** | *0.004* | | | | | | |
| Autumn overturn start | doy | **0.216\*** | *0.021* | | | | | | |
| Autumn overturn end | doy | **-0.122\*** | *0.012* | | | | | | |
| $T_s$ (summer mean) | °C | **0.024\*** | *0.012* | *0.015* | *0.071* | **0.018\*** | *0.033* | | |
| $h_{mix}$ (summer mean) | m | *-0.027* | *0.057* | | | | | | |



**Figures**

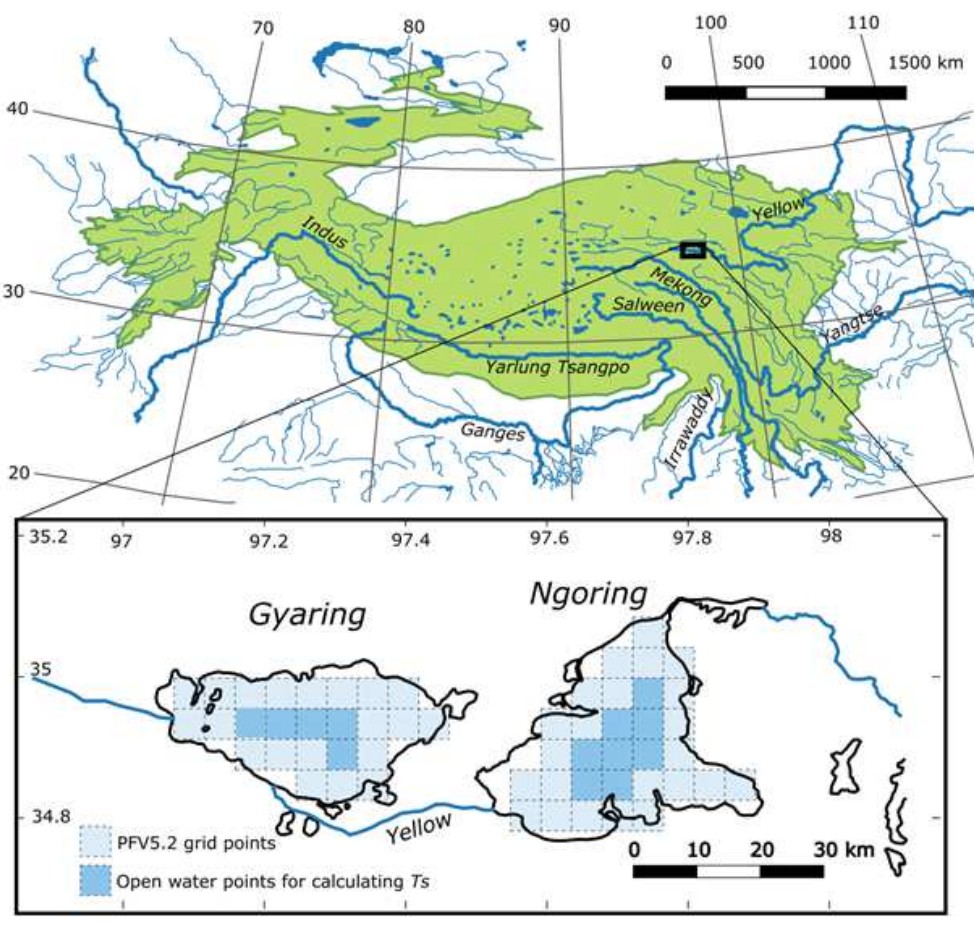

**Figure 1: Study site map and the points of the AVHRR Pathfiner grid used for estimation of the surface temperature from satellite**
5 **data. The green shaded region is the Tibetan Plateau, shown as land above 2000 m a.s.l.**





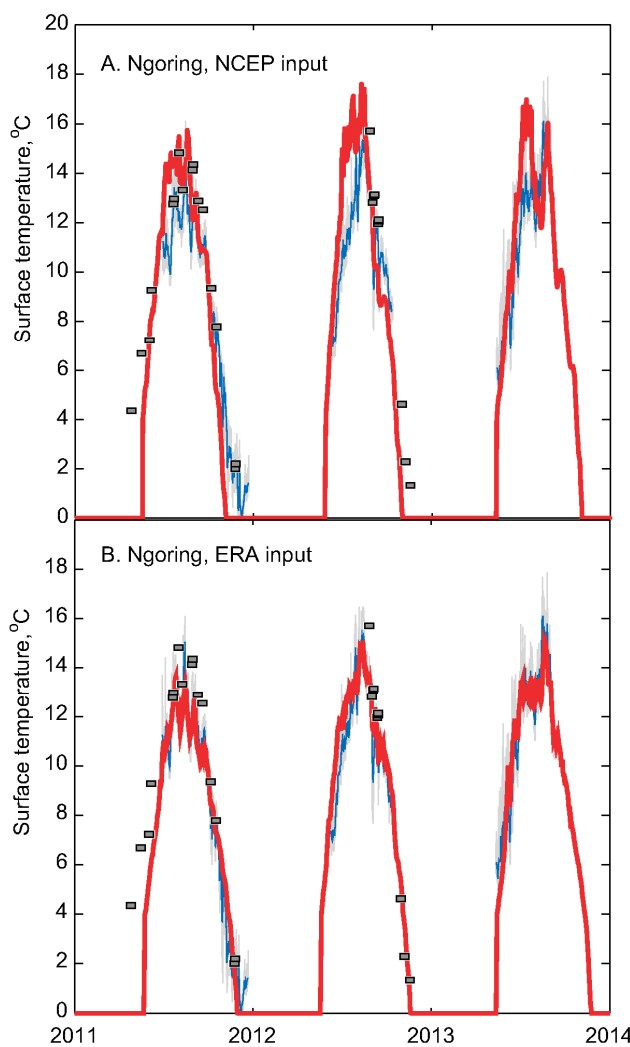

**Figure 2: Comparison of results from the Flake model forced by (a) NCEP/NCAR and (b) ERA-Interim reanalyses against the in-situ lake surface temperatures in Ngoring. Thick solid (red) lines are model predictions; thin solid (blue) lines are in situ bulk temperatures at 0.5 m depth; Half-tone (gray) lines are the skin surface temperatures determined from the upward long-wave radiation (Li et al. 2016); symbols are the available $T_s$ estimations from satellite data.**




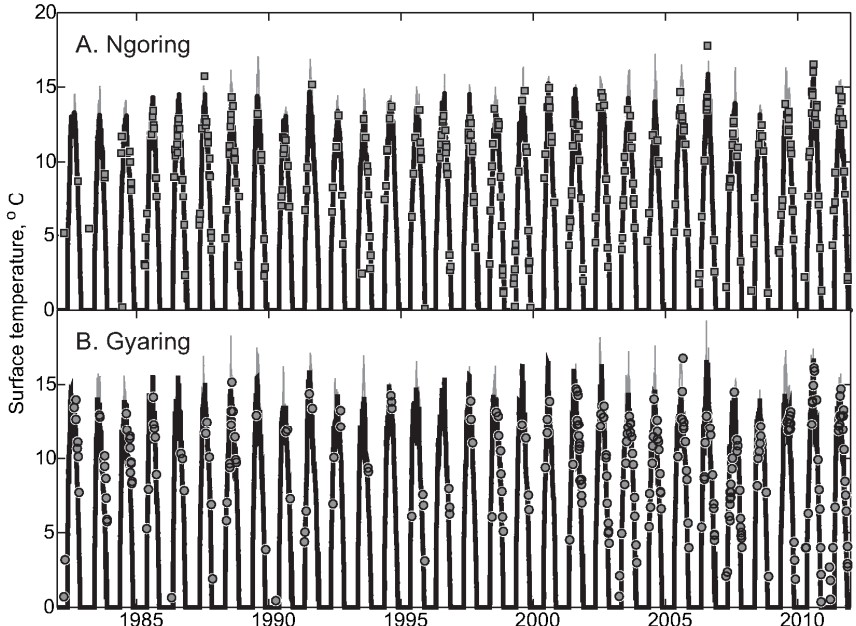

**Figure 3: Modeled lake surface temperatures (lines) and the Pathfinder SST data (symbols) for 1982-2012 in (a) Ngoring and (b) Gyaring. Thick lines are from the Flake model forced by the ERA-Interim reanalysis, thin gray lines are from NCEP/NCAR forcing.**





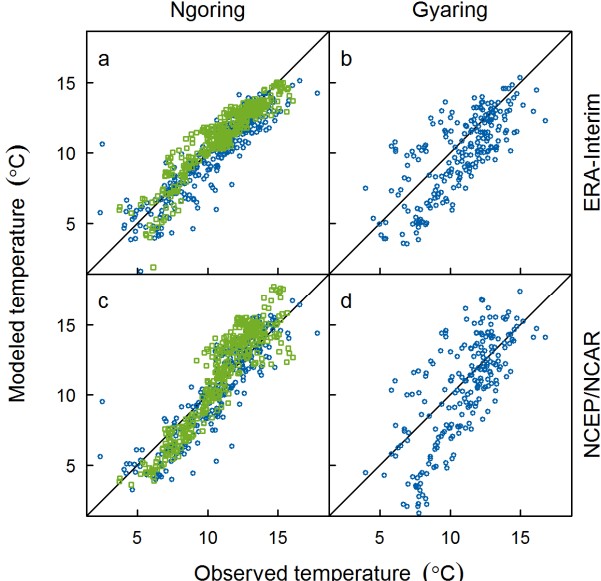

**Figure 4:** Comparison of observed and modeled lake surface temperatures in Ngoring (a, c) and Gyaring (b, d) forced by the ERA-Interim reanalysis (a, b) and the NCEP/NCAR reanalysis (c, d). Blue circles are derived from satellite data and green squares from in-situ measurements. Only values from May to October are shown.





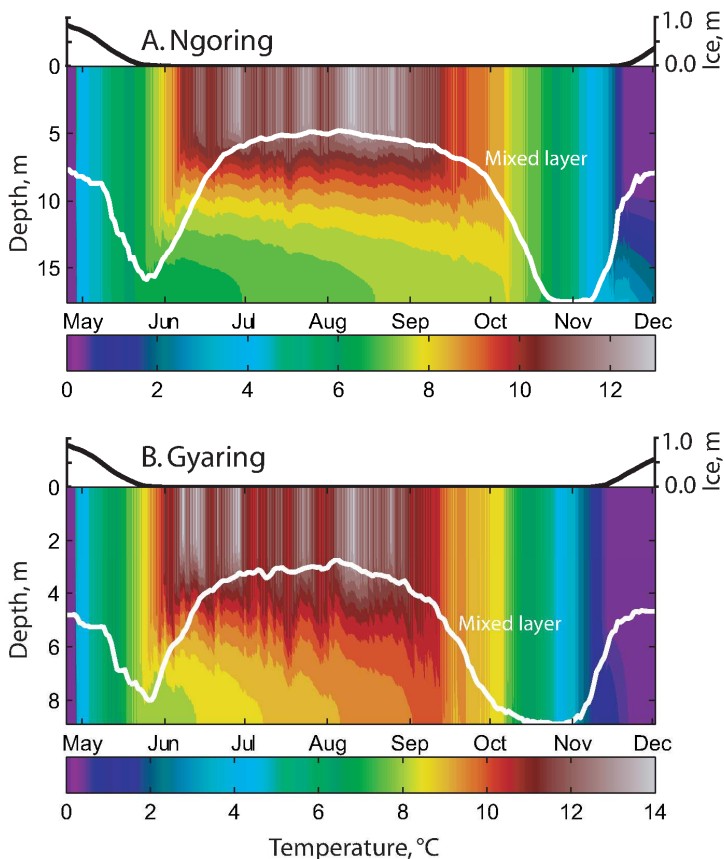

**Figure 5: Modeled seasonal thermal stratification pattern and ice cover development in (a) Ngoring and (b) Gyaring. The model was forced by the ERA-Interim reanalysis. Only values from May to December 1979 are shown. The black line is the ice cover thickness. The white line is the bottom of the surface mixed layer defined as the depth of the maximum vertical temperature gradient.**





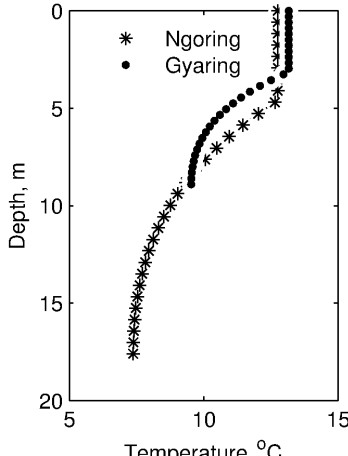

**Figure 6: Typical modelled vertical profiles of temperature in Ngoring (stars) and Gyaring (circles) in the middle of the summer stratification period (12 Aug 1979).**



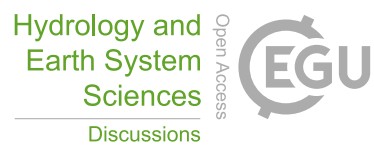

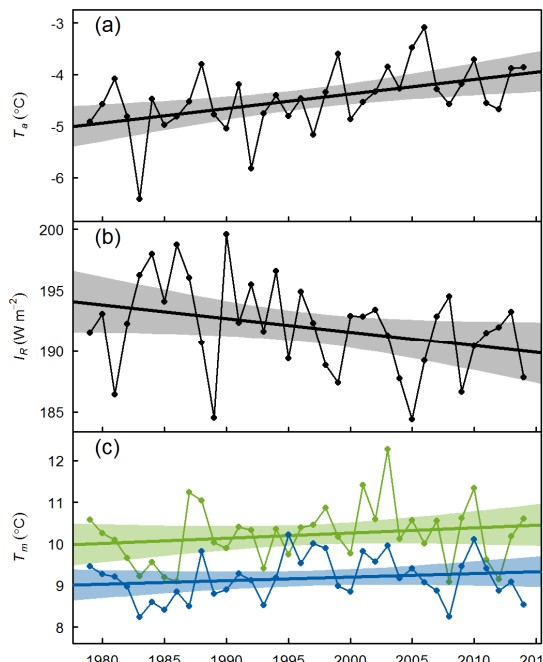

**Figure 7. Trends in annual mean air temperature (a) and solar radiation (b) from ERA-Interim, as well as trends in modeled mean summer lake temperature (c) in Ngoring (blue) and Gyaring (green).**