# Peer review of "Seasonal thermal regime and climatic trends in lakes of Tibetan Highlands"

_Hydrology and Earth System Sciences, 2016_

## Referee Comment (RC1) · Anonymous Referee #1 · 12 Jan 2017

Undoubtedly, this paper contains interesting new information and important results about poorly studied lake systems of Tibetan Plateau, as well as potential feedbacks between the hydrological regimes of small and medium size lakes on the one hand and the ongoing climate change on the other. The author's approach combines the use of in situ, reanalysis, and remote sensing data with numerical modeling, which adds robustness to the conclusions. The manuscript can be accepted for publication after a moderate revision, which, in my opinion, should address the following points:

General comments

1. The authors tend to extend their findings and conclusions onto all lakes of the Tibetan highlands, which are "thousands" distributed over a broad area, while factually they only investigated two individual lakes, situated close to each other. Therefore,

they should either confine their conclusions to the specific lakes Ngoring and Gyaring (perhaps, making it explicit even in the title of the article), or otherwise try to justify that these two lakes are somehow representative of the whole generality of Tibetan lakes. Is it possible to compile from the literature some kind of statistics on the principal characteristics of all Tibetan lakes (e.g., size, depth, salinity, turbidity, etc) from where it would follow that Ngoring and Gyaring are indeed "typical" examples?

2. The core result of the present study, as the authors put it, is the statement that on the climatic scale, "the effects of the air temperature increase on the lake heat budget were counteracted by the reduced radiative heating". While this concept is qualitatively clear and sounds very plausible, cannot it be made more quantitative? That is, can we estimate the changes which the air temperature trend of 0.5-0.7oK decade-1 evident in the reanalysis will produce in the heat flux through the lake surface and see whether the latter is sufficient to offset the decrease of solar irradiance at 3.8 W m-2 decade-1? If yes, this would make the author's hypothesis about the mutual compensation of the two trends in heat budget components much more grounded.

Specific comments

3. In my opinion, the description of the FLake model (section 2.3) should be detalized and made more concrete. It is not immediately clear to those unfamiliar with the model what does "self-similar representation of the temperature profile" mean (do I understand right that the shape of the profiles is prescribed and only the coefficients are fitted?) and what kind of prognostic entrainment equation was used. Of course, these details can be found in the references provided by the authors, but the reader should be able to get sufficient insight into the model without having to browse through the literature.

4. Figure 3 is barely legible at this scale. For example, the thin grey line is only distinguishable where the NCEP/NCAR-forced outcome exceeds the ERA-force one. I suggest simply deleting this figure, as it adds little, if anything, to the information

already presented in the tables.

5. Page 3, lines 31-32: "The mineralization of the lake water is low with a conductivity of ∼0.4 mS/cm-1" – can this be expressed in mg/l of salt content?

6. Same lines and page 4, first paragraph: Mineralization is specified only for Ngoring, while nothing is said about mineralization of Gyaring. Reciprocally, the Secchi depth is only specified for Gyaring, with no mention of that in Ngoring. This is confusing. Whenever possible, comparative characteristics should be given for both lakes.

7. Page 7, line 4: Looks like Fig. 4 is referred to before Fig. 3. Consider revising the numbering of figures.

———————————————————————

---

## Referee Comment (RC2) · Anonymous Referee #2 · 13 Jan 2017

This is a nicely written paper analyzing the climatic trends of 2 large lakes of the Tibetan highlands based on a combination of measurements and modeling. The modeled lake surface temperatures are compared to measured temperature in a first step (satellites and in situ measurements), and the climatic trends of the lake surface temperature in response to atmospheric trends are analyzed in a second step. The lake surface temperature is found to change only marginally, and the reason is attributed to counteracting effects of increasing air temperatures and decreasing radiation. I recommend minor revisions along the following points:

1- Though sound, the conclusion about the counteracting effects of increasing air temperature and decreasing radiation is a bit hypothetical in the present form. It would strengthen the conclusion if the authors quantified the expected effect of the decreasing radiation on the lake temperature, and the expected effect of the warming air tem-

perature on the heat budget component and to compare them.

2- a brief description of the Flake model would be beneficial to a smooth reading, since some detailed aspects of the physics included or not in the model are discussed in the model performance section. Also in the discussion of the model performance, the judgment words (appreciably better, etc. . .) could be backed up by basic statistics.

Minor comments: P1L21 and elsewhere: dampened should be damped

P1L22: I would recommend specifying "the modeled mean lake temperatures did not change".

P4L12: what is the time resolution of the bulk temperature? "All sensors . . .recorded the same temperatures": are you referring to diurnal averages?

P5L11: is there a mismatch in the figure? There seems to be more open water points than what the figure shows.

P8L28: it could be nice to add a small note on the trend in the satellite measurements as comparison (since there are several decades available).

P9L30: please provide some details about these bulk algorithms.

P11L26: Is the mixed layer thickness not an output of the model? If so, can the evolution in time of the modelled mixed layer thickness confirm the argument?

---

## Referee Comment (RC3) · Anonymous Referee #1 · 14 Feb 2017

I am satisfied with the revision and believe that the paper is now ready for publication.
* * *

---

## Author Comment (AC1) · 14 Feb 2017

We thank the two Referees for careful reading of the manuscript and positive evaluation of our work. Both reviewers made two identical suggestions on the improvement of the manuscript: (i) A request for some quantitative support of the counteracting effect on long-term lake temperature evolution of the increasing air temperature and decreasing solar radiation amount; (ii) a recommendation to provide some basic details on the model used in the study. We reply first to these two remarks and address the other minor remarks afterwards.

Our responses are marked in **bold face**, the outtakes from the revised manuscript are in *italic*.

**General reply 1. (Referee I, remark 2. Referee II, remark 1)** Justification of the canceling effect of the radiation decrease on the water temperature trend.

**We agree that a more detailed quantitative analysis of the long-term effects of the atmospheric trends would provide a stronger support to the suggested cause of the negligible lake warming over Tibet. We performed a set of model sensitivity experiments on this subject and described the outcomes in the Discussion part of the manuscript as follows:**

"*To quantify the counteracting effects of the increasing air temperature and the decreasing short-wave radiation we performed model runs using artificial short-wave radiation input constructed from ERA values (which had a significant negative trend) with the summer radiation trend removed. The trend was removed based on daily averaged values, and the diurnal cycle of solar radiation was subsequently restored to the detrended values based on daytime duration and the daily maxima of solar radiation. Additionally, a 'control' model run was performed with the radiation input reconstructed from daily averages using the same procedure, but without trend removal. The model results based on the modified 'control' radiation input did not differ significantly from those based on the original ERA input (not shown). In turn, the model driven by the detrended radiation input produced a positive trend in the mean summer surface temperature of 0.26 °C decade$^{-1}$ and the maximum summer surface temperature of 0.34 °C decade$^{-1}$. The trend in surface water temperatures was significant at > 99% level of confidence (p = 0.003), whereas the original ERA input produced no significant long-term increase of surface temperatures (0.15 °C decade$^{-1}$ at < 93% level of confidence, see Table 4). Hence, the short-wave radiation decrease did indeed cancel the effect of the air temperature increase on lake temperatures in ERA-driven scenarios. The hypothetical surface temperature trend 0.26 °C decade$^{-1}$ is still lower than the summer air temperature increase of 0.50 °C decade$^{-1}$, and is accompanied by significant changes in the summer stratification: a later breakdown of the stratification due to autumn cooling (1.8 days decade$^{-1}$, p = 0.034) and a shallower surface mixed layer depth (0.14 m decade$^{-1}$, p = 0.006). Several previous studies devoted to alpine or tropical lakes also reported lake temperature trends lower than the regional atmosphere warming (Livingstone 2003; Vollmer et al. 2005; Coats et al. 2006), while lowland temperate lakes are often found to be in a thermal equilibrium with the atmosphere (Arhonditsis et al. 2004; Danis et al. 2004; Kirillin 2010).*"

**2. In reply to the Referees' request (Referee I, Remark 3 and Referee II. Remark 2) we included a description of the model concept and additional references to the Methods":**

*For completeness, we provide here a short description of the model concept, following Kirillin (2010). For further details the reader is referred to the model manual (Mironov 2008). FLake is a one-dimensional model calculating the evolution of the horizontally averaged temperature in lakes. The basic principle underlying the model consists in splitting the water column into horizontal layers of variable depth, using physically-motivated assumptions about their properties, and subsequently integrating the governing equations over these layers. In this sense, FLake belongs to the family of*

bulk-models, of which the mixed layer models of the oceanic surface boundary layer *(Niiler and Kraus 1979)* are the most famous examples. In contrast to the traditional mixed-layer models, FLake additionally involves the "thermocline self-similarity" hypothesis to describe the lower stratified layer. The idea consists in adopting the thermocline thickness, $\Delta h(t)$, and the temperature jump across the thermocline, $\Delta T(t)$, as universal scales for the temperature profile, $T(t, z,)$ (Fig. 2). Using these scales, the dimensionless temperature, $\vartheta$, and the dimensionless vertical coordinate, $\zeta$, can be introduced as

$$\vartheta = \frac{T(t,z)-T_S(t)}{\Delta T(t)}, \; \zeta = \frac{z-h(t)}{\Delta h(t)} \;, \tag{1}$$

where $T_S$ is the temperature in the surface mixed layer and h is the mixed layer depth. The self-similarity of the temperature profile implies universality of the function $\vartheta(\zeta)$, or, in other words, a universal shape of the temperature profile within the stratified layer of all lakes. Several analytical expressions for $\vartheta(\zeta)$ have been proposed by various authors that all represent a similar thermocline shape supported by numerous observations (see the comprehensive review by Kirillin 2002). Assuming a two-layer lake temperature structure with a homogeneous upper layer and a self-similar stratified layer beneath, the vertical temperature profile across the entire water column can be described as (Fig. 2),

$$T(t,z) = \begin{cases} T_S(t) & at & 0 \leq z \leq h(t) \\ T_S(t) - \vartheta(\zeta)\Delta T(t) & at & h(t) \leq z \leq h(t) + \Delta h(t) \end{cases} . \tag{2}$$

The temperature profile represented by Eq. (2) does not depend explicitly on the vertical coordinate, which allows the heat transport equation to be integrated across the layers $0 < z < h$ and $h < z < (h+\Delta h)$, turning the partial differential equation for T(t,z) into a set of two ordinary differential equations (ODEs). The same self-similarity concept is used to describe the temperature structure of the thermally active upper layer of the bottom sediments (Golosov and Kirillin 2010) and of the ice cover (Mironov et al. 2012), so that the model, strictly speaking, is four-layered, with a unique self-similarity function $\vartheta(\zeta)$ for every layer: ice (if it exists), the mixed layer, the stratified layer, and the thermally-active sediment. Integration of the turbulent kinetic energy (TKE) equation across the upper mixed layer *(Niiler and Kraus 1979)* yields an additional ODE for the evolution of the mixed layer depth, h(t). The derivation procedure and the exact ODE expressions are given by Mironov *(2008)*. The surface boundary conditions enter the reformulated problem as the time-dependent functions for the surface heat and momentum fluxes; the latter is used in the TKE equation for estimating the impact of the wind-generated mixing on the mixed layer h(t) evolution. The short-wave solar radiation is not included into the boundary condition for the surface heat flux, but is treated as a separate heat source absorbed internally within the upper water column by the one-band exponential law for radiation decay. A separate model block calculates the surface heat and momentum fluxes from the standard meteorological variables and the water surface temperature at the previous time step *(Mironov 2008)*.

Referee 1.

1. The authors tend to extend their findings and conclusions onto all lakes of the Tibetan highlands, which are "thousands" distributed over a broad area, while factually they only investigated two

individual lakes, situated close to each other. Therefore, they should either confine their conclusions to the specific lakes Ngoring and Gyaring (perhaps, making it explicit even in the title of the article), or otherwise try to justify that these two lakes are somehow representative of the whole generality of Tibetan lakes. Is it possible to compile from the literature some kind of statistics on the principal characteristics of all Tibetan lakes (e.g., size, depth, salinity, turbidity, etc) from where it would follow that Ngoring and Gyaring are indeed "typical" examples?

**Reply: we did not claim the two investigated lake were 'typical' for the very versatile lake system of Tibet. They are the two largest freshwater lakes of the system. However, our methods of analysis, including modeling, allow to extend the majority of our findings on the freshwater lakes on the Plateau. We believe, no changes are necessary in the title. We clarified our point in the following text added to the discussion:**

*Our model experiments are well supported by the observational data from the largest freshwater lake of the Tibetan Plateau. They compare the dynamics of lakes with different depths and water transparencies under the same atmospheric forcing, characteristic for high-altitude conditions. The presented results provide the first detailed insight into the specific mechanisms of internal lake mixing and lake-atmosphere coupling, specific to the Tibetan lakes and extendable to other high-altitude hydrological systems. It should be noted that freshwater lakes comprise only about 10% of the Tibetan lake system (Jiang and Huang 2004), while ~75% of lakes have brackish or saline water (the salt content of the rest of lakes is unknown). The salt stratification can significantly contribute to the seasonal mixing regime as well as to the lake response to long-term changes in the atmospheric input. Effects of salinity, as well as lake physics during the ice-covered period, as discussed above, are the two major facets of lake dynamics for further clarification with regard to specific features of the Tibetan lake system.*

2. The core result of the present study, as the authors put it, is the statement that on the climatic scale, "the effects of the air temperature increase on the lake heat budget were counteracted by the reduced radiative heating". While this concept is qualitatively clear and sounds very plausible, cannot it be made more quantitative? That is, can we estimate the changes which the air temperature trend of 0.5-0.7oK decade-1 evident in the reanalysis will produce in the heat flux through the lake surface and see whether the latter is sufficient to offset the decrease of solar irradiance at 3.8 W m-2 decade-1? If yes, this would make the author's hypothesis about the mutual compensation of the two trends in heat budget components much more grounded.

**See General Reply 1.**

Specific comments

3. In my opinion, the description of the FLake model (section 2.3) should be detalized and made more concrete. It is not immediately clear to those unfamiliar with the model what does "self-similar representation of the temperature profile" mean (do I understand right that the shape of the profiles is prescribed and only the coefficients are fitted?) and what kind of prognostic entrainment equation was used. Of course, these details can be found in the references provided by the authors, but the reader should be able to get sufficient insight into the model without having to browse through the literature.

**See General Reply 2.**

4. Figure 3 is barely legible at this scale. For example, the thin grey line is only distinguishable where the NCEP/NCAR-forced outcome exceeds the ERA-force one. I suggest simply deleting this figure, as it adds little, if anything, to the information already presented in the tables.

**Reply: while we partially agree on the comment, we decided to keep the figure as demonstrating the available satellite data and providing a visual demonstration of the impossibility of trend simation based on the remote sensing only (see our reply to the comment on P8L28 of Reviewer II).**

5. Page 3, lines 31-32: "The mineralization of the lake water is low with a conductivity of ~0.4 mS/cm-1" – can this be expressed in mg/l of salt content?

**Reply: Since exact information on salt content is absent for both lakes, and the conductivity values are superfluous for the subsequent analysis we replaced the sentence with:**

*Also, both lakes are fresh, with salt content < 1 g l$^{-1}$ (Kar 2014), so that effects of salt on density stratification can be neglected in modeling.*

6. Same lines and page 4, first paragraph: Mineralization is specified only for Ngoring, while nothing is said about mineralization of Gyaring. Reciprocally, the Secchi depth is only specified for Gyaring, with no mention of that in Ngoring. This is confusing. Whenever possible, comparative characteristics should be given for both lakes.

**Reply: the Secchi depth has been given for both lakes. The text has been refined to make it unambiguous (Section 2.1 Study sites).**

7. Page 7, line 4: Looks like Fig. 4 is referred to before Fig. 3. Consider revising the numbering of figures.

**Reply: figures are re-ordered and the numbering is made consistent.**

Referee 2.

1- Though sound, the conclusion about the counteracting effects of increasing air temperature and decreasing radiation is a bit hypothetical in the present form. It would strengthen the conclusion if the authors quantified the expected effect of the decreas-ing radiation on the lake temperature, and the expected effect of the warming air temperature on the heat budget component and to compare them.

**See General Reply 1.**

2- a brief description of the Flake model would be beneficial to a smooth reading, since some detailed aspects of the physics included or not in the model are discussed in the model performance section. Also in the discussion of the model performance, the judgment words (appreciably better, etc. . .) could be backed up by basic statistics.

**See General Reply 2.**

Minor comments:

P1L21 and elsewhere: dampened should be damped

We replaced dampened with damped.

P1L22: I would recommend specifying "the modeled mean lake temperatures did not change".

**Reply: added.**

P4L12: what is the time resolution of the bulk temperature? "All sensors . . .recorded the same temperatures": are you referring to diurnal averages?

**Reply: Even at sub-diurnal time scales, the upper 0.6 m of the water column remained well-mixed within the 0.1°C accuracy of the sensors.**

P5L11: is there a mismatch in the figure? There seems to be more open water points than what the figure shows.

**Reply: one of the earlier versions of the figure missed one open water open water point in Gyaring. There are 5 and 8 open points for Gyaring and Ngoring respectively.**

P8L28: it could be nice to add a small note on the trend in the satellite measurements as comparison (since there are several decades available).

**Reply: we considered this option during the analysis but had to abandon it. An explanation is added to the revised manuscript as:**

*The satellite data on lake surface temperatures, despite the coverage of several decades, were found inappropriate for reliable trend estimation due to numerous data gaps of several months, exceeding the half-period of the seasonal variations (Fig. 3)*

P9L30: please provide some details about these bulk algorithms.

**Reply: We extended the sentence as following:**

*Taking into account strong constraints on the computational costs of the lake parameterization schemes, a remarkable improvement in regional climate models can be achieved by extending the lake modules to include one of the bulk algorithms developed earlier for convective mixing based on integration across the mixed layer of the equation for mixing energy production with solar radiation as a volumetric energy source (Mironov et al. 2002; Oveisy and Boegman 2014).*

P11L26: Is the mixed layer thickness not an output of the model? If so, can the evolution in time of the modelled mixed layer thickness confirm the argument?

**Reply: the argument is based on the modeling results on the mixed layer trends (last paragraph of Results).**